# Adequate Urinary Iodine Concentration among Infants in the Inland Area of Norway

**DOI:** 10.3390/nu13061826

**Published:** 2021-05-27

**Authors:** Kjersti Sletten Bakken, Tonje Eiane Aarsland, Synne Groufh-Jacobsen, Beate Stokke Solvik, Elin Lovise Folven Gjengedal, Sigrun Henjum, Tor Arne Strand

**Affiliations:** 1Women’s Clinic at Lillehammer Hospital, Innlandet Hospital Trust, 2609 Lillehammer, Norway; tonje.aarsland@sykehuset-innlandet.no (T.E.A.); beate.stokke.solvik@sykehuset-innlandet.no (B.S.S.); 2Center of International Health, Faculty of Medicine, University of Bergen, P.O. Box 7804, 5020 Bergen, Norway; tors@me.com; 3Department of Nutrition and Public Health, Faculty of Health and Sport Sciences, University of Agder, Universitetsveien 25, 4630 Kristiansand, Norway; synne.groufh.jacobsen@uia.no; 4Faculty of Environmental Sciences and Natural Resource Management, Norwegian University of Life Sciences, 1432 Aas, Norway; elin.gjengedal@nmbu.no; 5Department of Nursing and Health Promotion, Faculty of Health Sciences, OsloMet–Oslo Metropolitan University, 0130 Oslo, Norway; shenjum@oslomet.no; 6Department of Research, Innlandet Hospital Trust, 2609 Lillehammer, Norway

**Keywords:** infants, iodine, knowledge, urinary iodine concentration, UIC, iodine intake, inland area, Norway

## Abstract

Considering the importance of iodine to support optimal growth and neurological development of the brain and central nervous system, this study aimed to assess and evaluate iodine status in Norwegian infants. We collected data on dietary intake of iodine, iodine knowledge in mothers, and assessed iodine concentration in mother’s breast milk and in infant’s urine in a cross-sectional study at two public healthcare clinics in the inland area of Norway. In the 130 mother–infant pairs, the estimated infant 24-h median iodine intake was 50 (IQR 31, 78) µg/day. The median infant urinary iodine concentration (UIC) was 146 (IQR 93, 250) µg/L and within the recommended median defined by the World Health Organization for this age group. Weaned infants had a higher UIC [210 (IQR 130, 330) µg/L] than exclusively breastfed infants [130 (IQR 78, 210) µg/L] and partially breastfed infants [135 (IQR 89, 250) µg/L], which suggest that the dietary data obtained in this study did not capture the accurate iodine intake of the included infants. The iodine status of infants in the inland area of Norway seemed adequate. Weaned infants had higher UIC compared to breastfed infants, suggesting early access and consumption of other sources of iodine in addition to breast milk.

## 1. Introduction

Iodine is an essential mineral required for the synthesis of thyroid hormones (THs). In the first phases of life, optimal levels of THs are critical for normal growth and neurological development of the brain and central nervous system [1]. Iodine excess may also result in thyroid dysfunction, but the evidence for health effects of excess is more limited than of deficiency [2]. Iodine deficiency (ID) in the fetus and infant is the most common cause of preventable brain damage globally [3]. A systematic review from 2013 on the effect of ID on mental development in children 5 years and younger showed that even mild ID could influence school performance, intellectual ability, and work capacity of children [4]. However, a recent review concluded that there is insufficient evidence to support recommendations for iodine supplementation in areas of mild- or moderate deficiency of iodine [5]. Despite the role of iodine in growth and development, we have little knowledge of infant’s iodine status in Norway.

Infant sources of iodine come exclusively from breast milk or formula during the first 4–6 months of life and thereafter from breast milk, formula, and complementary foods. The iodine concentration in breast milk reflects maternal iodine intake and probably status, which in turn is affected by the maternal intake of dietary iodine and/or supplements. The main sources of iodine in the Norwegian diet are milk and seafood, mainly lean fish such as cod, saithe, and haddock, constituting approximately 55 and 20% of the dietary iodine intake, respectively [6]. Severe ID was widespread in Norway 100 years ago, especially in the inland area where the seafood intake was low [7]. In the 1950s, iodine was added to cow fodder to improve animal health [8]. Due to the high transfer of iodine to milk and high consumption of milk and dairy products, the iodine intake in the population increased. However, a decline in the intake of these foods over the last decade has contributed to inadequate iodine intake in several groups of the population, especially among women of childbearing age [9,10,11,12]. Iodine is excreted in breast milk during lactation, and breastfed infants rely on an adequate supply through the breast milk to cover their production of thyroid hormones [13]. As the breastfeeding prevalence in Norway is high, a large proportion of infants may be at increased risk of not reaching their iodine needs due to inadequate maternal iodine intake. The World Health Organization (WHO) recommends a daily iodine intake of 90 µg/day for children younger than two years of age [14], while in the Nordic countries a daily iodine intake of 50–70 µg/day is recommended [15].

As severe ID is no longer seen in Norway, the primary concern is the consequences of mild-to-moderate ID. Globally, the main strategy to eliminate ID is iodization of salt, which is not implemented in Norway. A risk-benefit analysis by the Norwegian Scientific Committee for Food and Environment on the health consequences of iodization of household salt and salt used in bread was published recently. This report concluded that these measures could benefit groups such as youths and women of childbearing age but simultaneously put one- and two-year-old’s at risk of excess iodine intake [16]. The report highlights the need for more data on small children’s iodine status in Norway. A recent study confirmed inadequate iodine intake and insufficient iodine status among lactating women in the inland area of Norway [17]. This study aimed to assess and evaluate iodine status in infants resident in the same area.

## 2. Materials and Methods

### 2.1. Study Population

During October–December 2018, mother-infant pairs were recruited at two public healthcare clinics, in the cities Lillehammer and Gjøvik. Mothers and infants aged 0–12 months with appointments at the healthcare clinics were all invited to participate in the study. The inclusion criteria were (1) mothers were able to read and write in Norwegian, (2) healthy infant, no known metabolic or congenital chronically illness in the infant that could affect cognitive development, and (3) possibility to collect urine and blood sample of the infant.

A total of 204 mothers were invited to participate, and of these 151 (74%) mother–infant pairs signed informed consent at the public healthcare clinics. Eleven participants withdraw from the study and four were lost to follow-up (Figure 1). In addition, six mother-infant pairs were excluded from this analysis due to use of thyroid medication. Completed questionnaires and urine samples were available for 130 infants. Figure 1 shows the study recruitment.

### 2.2. Ethical Considerations

The Regional Committee for Medical and Health Research Ethics approved the study (2018/1230/REC South East). All women provided written informed consent. All materials were handled de-identified.

### 2.3. Collection of Breast Milk and Urine Samples

Every mother was informed about the study purpose, and the ones that consent to participate were instructed to collect one spot breast milk sample (5 mL) in the morning before breastfeeding and a second spot breast milk sample (5 mL) in the afternoon after breastfeeding. Both breast milk samples were obtained by manual expression into the same labelled 50 mL polypropylene (pp) centrifuge tube (Sarstedt, Nümbrecht, Germany). Moreover, the mothers were instructed to collect one spot urine sample (5 mL) from the infant in the morning using Sterisets Urine collection packs 310,019 (Sterisets International BV, Oss The Netherlands) containing one syringe (5 mL), one specimen container (20 mL), two uricol collection pads (21 cm × 7 cm), and an instruction leaflet. To collect the baby’s urine, a pad was placed inside the disposable diaper and checked every 5 min until the pad was wet with urine but not soiled by feces. Then, the pad was placed on a flat surface, and the urine was transferred into the sterile container. An alternative method to extract the urine was to cut the pad open, put the wet cotton into the opened syringe and press the urine into the container. In summary, once during the study period, each mother donated a repeated spot breast milk sample in one container (10 mL) and one spot infant urine sample (5 mL), all samples collected on the same day. The containers were kept refrigerated until the sample was transferred to a −70 °C freezer and later transported on dry ice for analysis.

### 2.4. Chemical Analyses

The analysis of the mother’s breast milk and the infant urine to determine the concentration of iodine was performed at the Norwegian University of Life Sciences, Faculty of Environmental Sciences and Natural Resource management. The frozen urine was thawed and aliquoted into 15 mL pp centrifuge tubes (Sarstedt, Nümbrecht, Germany) using a 100–5000 µL electronic pipette (Biohit, Helsinki, Finland). In detail, an aliquot of 1.00 mL of urine was diluted to 10.0 mL with an alkaline mixture (BENT), containing 4% (weight (w)/volume (V)) 1-Butanol, 0.1% (*w/v*) H4EDTA, 2% (*w/v*) NH4OH, and 0.1% (*w/v*) TritonTM X-100 (Millipore, Burlington, MA, USA). To avoid precipitation of struvite (MgNH4PO4 × 6H2O) in the urine, the concentration of NH4OH in BENT was set to 2%.

The breast milk samples were thawed and heated to 37 °C in a heating cabinet, homogenized, and then prepared by dilution in an alkaline solution (BENT), following the same procedure as the one used for the urine samples, except that the concentration of NH4OH in BENT was increased to 5% (*w/v*). A conformance test between weight and volume of breast milk limited the concentration of iodine to two significant figures.

Standard Reference Materials (SRM) and method blank samples were prepared in the same manner as the respective sample matrices. Deionized water (>18 MΩ) and reagents of analytical grade or better were used throughout. The quantification of iodine in urine and breastmilk was performed by means of an Agilent 8900 ICP-QQQ (Triple Quadrupole Inductively Coupled Plasma Mass Spectrometer; Agilent Technologies, Hachioji, Japan) using oxygen reaction mode. Iodine was quantified on mass 127. To correct for non-spectral interferences, ^129^I was used.

In breast milk, the limit of detection (LOD) was 0.04 µg/L and the limit of quantification (LOQ) was 0.14 µg/L. With regard to the infant urine, LOD was 0.1 µg/L and LOQ was 0.44 µg/L. The LOD and LOQ were calculated by multiplying the standard deviation of five method blank samples that followed each treatment by three and ten, respectively. The measurement repeatability was 1.6% with respect to both urine and breast milk. To ensure methodological traceability and to check for accuracy, SRM were analyzed concurrently with the sample matrices. Allowing for a coverage factor k = 2, corresponding to a level of confidence of about 95%, our results were within the recommended values issued for the Seronorm™ (Oslo, Norway) Trace Elements Urine L-1, SeronormTM Trace Elements Urine L-2, and the European Reference Materials ERM^®^-BD 150 (Geel, Belgium) and ERM^®^-BD 151 Skimmed milk powders.

A performance test of the sampling method of infant urine showed no significant change in analyte; thus, the sampling method is considered reliable [18].

### 2.5. Infant Iodine Intake from Food and Supplements

To calculate the 24-h iodine intake in exclusively breastfed infants, reference values for human milk intakes by infants age in high-income countries [19] were multiplied with the mother’s individual breast milk iodine concentration (BMIC). For example, the breast milk intake of exclusively breastfed 1-month-olds is 0.699 L per 24 h. The estimated intake for a particular 1-month-old baby is accordingly 0.699 × BMIC µg/L. The BMIC data can only be used to estimate one 24-h iodine intake in breastfed infants as the mothers provided breast milk samples from one single day. Similarly, for partially breastfed infants, the 24-h iodine intake was calculated by combining the calculated iodine intake from breast milk (the intake of partially breastfed infants at 1 month is 0.611 L) with the calculated iodine intake from foods as estimated by the on-site 24-h recall.

The 24-h recall captured the infant’s intake of iodine-containing food items, reported by the mothers. The iodine intakes of partially breastfed- and weaned infants were calculated by using the 24-h recall data and the applicable iodine concentration according to the Norwegian Food Composition Table [20].

Habitual iodine intake in the infants who eat solid food was calculated based on a food frequency questionnaire (FFQ) assessing infant’s food intake the past two weeks. The questionnaire was developed based on a modified version used in previous studies [21]. The infant FFQ assessed the intake of bread, yoghurt, fatty fish, lean fish, fish products, iodine enriched smoothies, processed baby foods (dinner), eggs, cheese, water, and industrially produced porridge. The frequency alternatives ranged from never/rarely, more rarely than weakly, 1–3 times a week, 4–6 times a week, 1–2 times a day, 3–4 times a day, to 5 times a day or more. The reported daily consumption frequencies for each food item in the infant FFQ were multiplied with portion sizes and iodine concentrations applicable for each food item using The Norwegian Food Composition Table [20]. The habitual intake among partially breastfed infants only covers calculations from food, not the additional amount of iodine they get from breast milk.

### 2.6. Participant Characteristics and Independent Variables

We recorded the participants’ age, both maternal (in years) and infant (in weeks). We also recorded maternal educational level (in categories; <12 years, 12 years, 1–4 years college/university, and >4 years college/university), maternal height and weight, and calculated body mass index (BMI, kg/m^2^, categorized into; underweight [<18.5], normal weight [18.5–24.9], overweight [25–29.9] and obese [>30]), infant gender (boy or girl) and infant breastfeeding status (categorized into; weaned, partially, and exclusively). Information on maternal use of iodine-containing supplements were collected from the 24-h recall and reflects maternal use of the supplement at the time of breast milk collection.

The maternal iodine knowledge score was calculated using a validated questionnaire including six questions [22]. In the current study, we used three of the questions to calculate a total iodine knowledge score. These questions were: (1) What are the most important dietary sources of iodine? (2) Why is iodine important? and (3) What do you know about the current iodine status among pregnant women in Norway. The questions had multiple answer alternatives, whereas some were correct, and some were incorrect. A correct answer gave 2 points, if they correctly identified an incorrect answer they got 1 point, and a wrong answer gave 0 points. Those who answered “I don’t know” did not get 1 point, even if they “correctly identify an incorrect answer”. The total knowledge score ranged from 0–26 points and was categorized into four categories of knowledge scores: poor (0–5 points), low (6–11 points), medium (12–19 points) and high (20–26 points).

### 2.7. Statistical Analyses

Statistical analyses were performed using Stata/SE 16.1 (StataCorp, College Station, TX, USA). Skewness and kurtosis test for normality was used to test whether the dependent variable UIC was normally distributed. The correlation between variables were evaluated using Spearman correlation matrix. We used kernel regression to assess the univariate associations between the dependent variable infant UIC and the independent variables. We computed bootstrap standard errors and percentile confidence intervals with 100 replications by the npregress kernel command (non-parametric kernel regression).

## 3. Results

### 3.1. General Characteristics

Table 1 summarizes the background information of the infants and their mothers. Only three mothers reported to be either vegetarian or vegan, and only two mothers had never breastfed their infant. None of the infants consumed iodine-containing supplements; however, 23% of the mothers used iodine-containing supplements during the last 24 h.

### 3.2. Dietary Iodine Intake

The habitual, as well as the recent 24-h dietary iodine intakes are given in Table 2. In total, the estimated 24-h median (IQR) iodine intake was 50 (31, 78) µg/day. Eighty-eight (57.5%) of the infants had an iodine intake below the Nordic recommendations (50 µg/day for infants aged 6–11 months). The weaned infants seemed to have a lower dietary iodine intake compared to the breastfed infants (Table 2). Only 4.7% of these weaned infants were estimated to reach the intake recommendation of 50 µg iodine a day, whereas 54.7% of the exclusively breastfed infants had an estimated iodine intake above this recommendation. Seventeen (15.5%) of the weaned infants had a habitual iodine intake below the Nordic recommendations. Habitual iodine intake was not calculated for the exclusively breastfed infants, as two spot samples of breastmilk collected on the same day do not cover the day-to-day variation in breast milk iodine content.

### 3.3. Urinary Iodine Concentration

The median UIC (IQR) was 146 (93, 250) µg/L. A total of 44 (33.6%) of the infants had a UIC within the suggested optimal UIC range (100–199 µg/L). The exclusively breastfed and partially breastfed infants had a median UIC within the range, while the weaned infants had a median UIC slightly above (Table 2 and Figure 2). The association between breastfeeding status and UIC was significant (Table 3). Infants who were exclusively and partially breastfed had on average 79.6 µg/L (*p* = 0.006) and 70.2 µg/L (*p* = 0.014) lower UIC, respectively, compared to weaned infants. We did not identify other maternal or infant characteristics associated with the UIC (Table 3).

For the exclusively breastfed infants, almost half of the variation in UIC was explained by the maternal BMIC (adjusted R2 = 0.45). We found a linear relationship between infant UIC and BMIC (Figure 3). However, there was no significant correlation between infant UIC and the estimated 24-h dietary iodine intake (rs = 0.122, *p* = 0.166), which may be evidence of inaccuracies in the estimation of the dietary iodine intake.

## 4. Discussion

This is the first study to present data on iodine status and iodine intake in infants in the inland area of Norway. We found that the median UIC (146 µg/L) was indicative of sufficient iodine status according to the current WHO median of 100 µg/L in children under two years of age [14]. The median assumes a daily urine volume of 500 mL and 92% iodine bioavailability. A median UIC of 100 µg/L would extrapolate to a mean daily iodine intake of 55 µg, lower than the WHO recommendation of 90 µg/day iodine [23]. The disagreement in recommendations by the WHO makes evaluation of iodine status in infants difficult. There may therefore be a need for a review of the UIC recommendations by the WHO in small children using associations with biomarkers of iodine status, as the current one may not reflect the true metabolic iodine status in infants and young children. The thyroid-derived protein thyroglobulin (Tg) is increasingly being used as an index of iodine status in the population. In school-aged children, Tg is a sensitive indicator of both low and excess iodine intake [24]. However, validated reference ranges are needed for the determination of iodine sufficiency in infants and young children.

The infant median iodine intake based on 24-h recalls was 50 µg/day, which is below the WHO intake recommendation [14] but in accordance with the Nordic recommendation of 50–70 µg/day [15]. Since there are many challenges related to dietary assessment, our iodine intake estimations may be more accurate for exclusively breastfed infants than for partially breastfed and weaned infants.

Despite significant improvements over the past decades in the global iodine status through salt iodization, ID remains a concern in several countries, particularly in Europe. However, there is limited information on UIC in infants in European countries. Iodine status in infants is usually not monitored in countries with established salt iodization programs where the general population has adequate iodine intake. Breastfed infants may benefit from iodized salt through breastmilk, but the salt content in complementary foods and industrial baby foods is generally low [25]. It is estimated that up to half of all newborns in Europe may be at risk of restrictions to their cognitive potential due to ID [26]. Studies from other areas of Norway have been conducted, but most previous studies have assessed iodine status in subpopulations or in children at an older age. A study carried out in children with cow´s milk protein allergy, two years and younger, showed a median UIC of 159 µg/L [27]. The Little in Norway study (LiN) undertaken between 2011–2014, which included 18 months old infants from all health regions in Norway, found a median UIC of 129 µg/L, and the authors concluded that 59% had an adequate iodine status [28]. The iodine exposure in two-year-olds was estimated from the study Småbarnskost 3 conducted in 2019, which found a mean iodine intake of 138 µg/day and 128 µg/day respectively with and without iodine supplementation [29]. Småbarnskost 3 was based on semi-quantitative FFQs; providing a better estimate of the individual iodine intake than the UIC, but may overestimate the food intake and the intake of energy-related nutrients [30]. A validation study confirmed that the FFQ used in the nationwide survey overestimated the intake of energy and most nutrients [31].

Only one of the variables we tested, breastfeeding status, were associated with infant UIC. We did not find that increased maternal iodine knowledge score was associated with increased infant UIC. Few other studies have explored this. However, another Norwegian study among pregnant and lactating women found no association between the participants’ knowledge scores and iodine status [22]. In contrast, a study among young female students in Norway found that UIC was lower in women with high iodine knowledge compared to those with low knowledge score [12]. The majority of mothers in the current study (79%) had medium to high knowledge score, but the relationship between iodine knowledge score and UIC remains unclear.

We found significant differences in UIC according to breastfeeding status. All three groups had a median UIC above the median of 100 µg/L. There was a significantly higher prevalence of UIC values below the median among the exclusively and partially breastfed infants compared to the weaned infants. These variations may be explained by different urinary volumes between the breastfed and the weaned infants. It is therefore unfortunate that we did not measure creatinine in the urine samples. However, we believe that most of the differences in UIC between the groups can be explained by dietary factors related to breast milk, as there were negligible differences in UIC between the exclusively breastfed and partially breastfed infants. It is reasonable to believe that urine volume is also altered when going from being exclusively breastfed to being partially breastfed. A former publication from the present study confirmed mild to moderate ID and inadequate iodine intake among lactating women [17]. Since infants have limited iodine stores, an adequate BMIC is required to ensure adequate intake of the infant [32]. We also found that the BMIC explained almost half of the variation in UIC of the breastfed infants. The highest median UIC was among the weaned infants. However, this group seemed to have the lowest estimated iodine intake based on 24-h recall, less than half the intake compared to both the partially breastfed and the exclusively breastfed infants. The median UIC of 146 µg/L extrapolates to a median iodine intake of 67 µg/day, assuming the aforementioned conditions of urinary volume and iodine bioavailability. The low calculated iodine intake and the satisfactory UIC suggest that the dietary data did not capture the accurate iodine intake of the included infants. This was supported by the poor correlation between UIC and estimated iodine intake from the 24-h dietary intake.

As mentioned, there are many challenges related to dietary assessment, particularly in infants and young children. Dietary assessment methods comprise both under- and over-reporting [33] and thus introduce substantial errors into the calculation of both energy and specific nutrients such as iodine. The quality of dietary data is highly dependent on the caregiver´s ability to report the type and amount of food the young child has eaten. In this study, the infant food intake was reported by the mothers, although some of the children were with their fathers in the daytime. Obtaining accurate data may be further complicated if persons other than the parents are responsible for feeding. Some of the oldest infants in this study had daytime care and thus knowledge of the foods consumed was limited. Other challenges in dietary assessment in this age group are estimating the proportion of food spillage and wastage, particularly during and after the shift to self-feeding. These are possible explanations for the low estimated iodine intake in this study, particularly in the weaned infants. The estimated median habitual iodine intake in the weaned infants was 34 µg/day. On the other hand, calculating the 24-h iodine intake from breast milk may have led to an overestimation of the iodine intake in partially and exclusively breastfed infants. The reference values of breast milk intake used in this study may deviate from the volume consumed, and using these standards may thus have under- or overreported the iodine intake in the breastfed infants. Since the habitual intake among breastfed infants did not cover the iodine contribution from breast milk, this calculation may be underestimated.

Since only two maternal spot samples of breast milk were collected during the same day, the BMIC data could not be used to estimate the habitual iodine intake in the breastfed infants. Questions covering habitual use of caviar and whey cheese spread in infants were not included in the FFQ. However, the 24-h recall showed a low consumption of these foods. An additional challenge in calculating the iodine intake is the variability in the iodine concentration of many foods, particularly seafood. A study assessing the iodine content of six fish species in Norway found the iodine concentration in cod to range between 22–720 µg [6]. This may have under- or overestimated our iodine intake estimates as food composition tables do not account for natural variations in food.

An important strength of this study is the use of two spot samples of breast milk from the same day to estimate the iodine intake in exclusively breastfed infants. As BMIC vary throughout the day in response to recent iodine intake, two samples per mother provide a more accurate level of the BMIC. In addition to the already mentioned limitations related to dietary assessment, we had a relatively small sample size, particularly in the group of weaned infants. This can be partly explained by the age of our study group (0–12 months) and the recommendation to breastfeed for the entire first year. Another limitation is that the participants are not representative of the Norwegian population as they were recruited from only two public healthcare clinics in the inland area of Norway.

The National Council of Nutrition strongly advice national authorities to introduce mandatory iodization of 20 µg/g household salt and salt used in bread and bakery products [30]. They consider a low risk of excessive iodine intake in children and other groups of the population with the proposed enrichment. That report was not based on a systematic review or meta-analyses such as the risk-benefit report by The Norwegian Scientific Committee for Food and Environment, which considers all the relevant studies on iodine status in the population in Norway [16]. However, this study was a contribution to the knowledge need on iodine status in this population group.

## 5. Conclusions

The infants in the inland area of Norway had adequate iodine status evident by the median UIC. Weaned infants had higher UIC compared to breastfed infants, suggesting early access and consumption of other sources of iodine in addition to breastmilk. Breastfed infants had lower UIC compared to weaned infants. The results of this study add important information to the sparse literature on UIC and iodine intake in infants in Norway.

## Figures and Tables

**Figure 1 nutrients-13-01826-f001:**
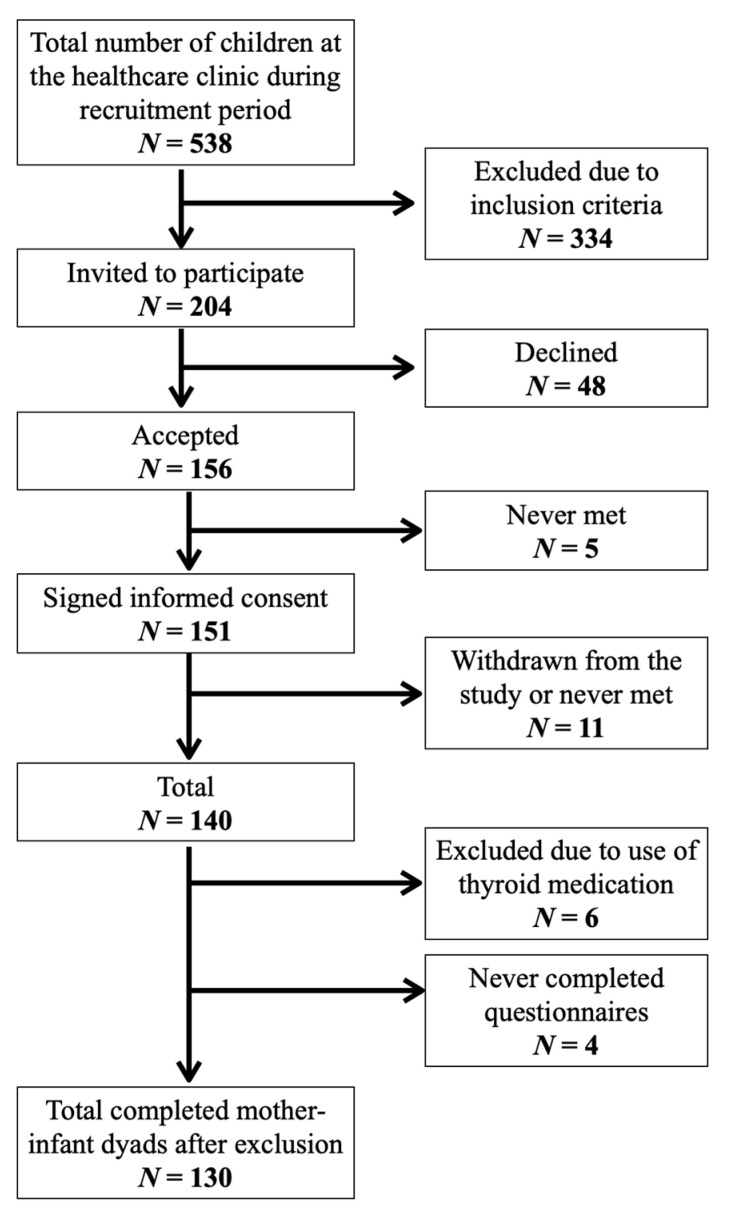
Flowchart of study recruitment and completion.

**Figure 2 nutrients-13-01826-f002:**
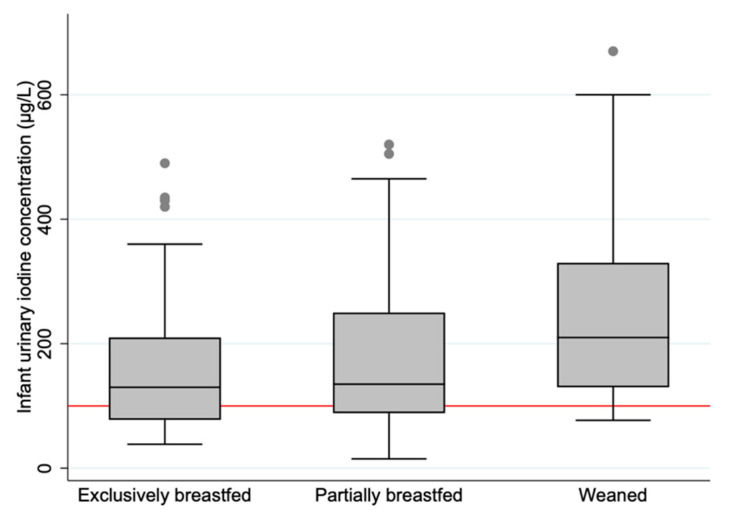
Boxplot displaying infant urinary iodine concentration (µg/L) in the three categories of breastfeeding status. The horizontal black line indicates the median; the boxes indicate the interquartile range (IQR); the whiskers represent observations within 1.5 times the IQR. The red line shows the recommended median value for sufficient iodine concentration in urine (≥100 µg/L).

**Figure 3 nutrients-13-01826-f003:**
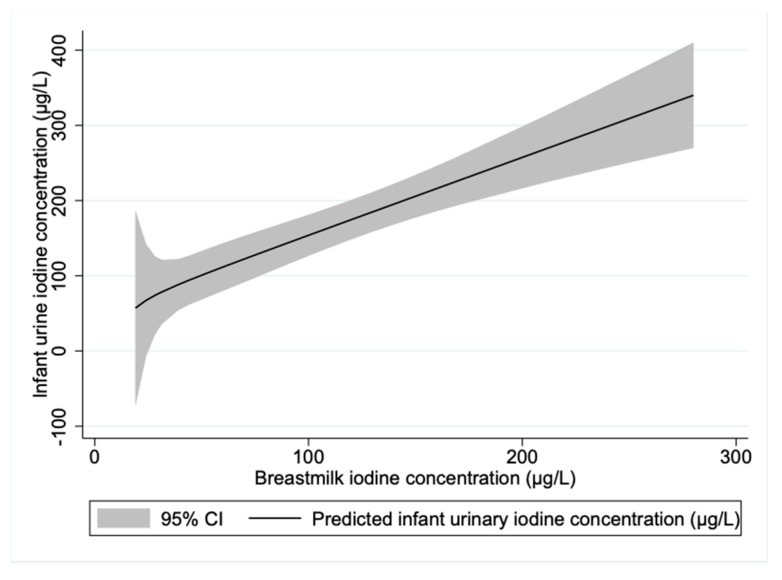
Prediction plot of infant urinary iodine concentration by maternal breastmilk iodine concentration for exclusively breastfed infants. *n* = 56.

**Table 1 nutrients-13-01826-t001:** Characteristics of mother–infant dyads (*n* = 130).

Characteristic	Categories	*n* (%) ^a^
Maternal age, mean (SD)		31.5 (4.6)
Maternal educational level	<12 years	6 (5)
	12 years	17 (13)
	1–4 years college/university	52 (40)
	>4 years college/university	55 (42)
Maternal BMI, kg/m^2^	<18.5 (Underweight)	4 (3)
	18.5–24.9 (Normal weight)	81 (62)
	25–29.9 (Overweight)	30 (23)
	>30 (Obese)	15 (12)
Maternal iodine knowledge score	Poor (0–5)	10 (8)
	Low (6–11)	17 (13)
	Medium (12–19)	69 (53)
	High (20–26)	34 (26)
Maternal use of iodine-containing supplement last 24-h	Yes	30 (23)
Infant age in weeks, median (min–max)		22 (1–5)
Infant gender	Boy	69 (53)
Breastfeeding status ^b^	Weaned	28 (22)
	Partially	46 (35)
	Exclusively	56 (43)

^a^ Numbers are presented as *n* (%) if not indicated otherwise. ^b^ Exclusively breastfed infants were defined as infants who received breast milk (including milk expressed) only, and no other liquids, solid foods or water was given, except drops of vitamins or minerals.

**Table 2 nutrients-13-01826-t002:** Urinary iodine concentration (µg/L) and calculated iodine intake (µg/day) for the infants (*n* = 130). Numbers are presented as median (IQR).

	Total *n* = 130	Exclusively Breastfed *n* = 56	Partially Breastfed *n* = 46	Weaned *n* = 28
Infant urinary iodine concentration, µg/L	146 (93, 250)	130 (78, 210)	135 (89, 250)	210 (130, 330)
Total habitual iodine intake, µg/day	-	-	21 (6, 37) ^a^	34 (14, 87)
Total 24-h iodine intake, µg/day	50 (31, 78)	66 (44, 107)	57 (35, 77)	25 (13, 39)

^a^ Includes iodine intake from solid food only, not breastmilk. Habitual intake in exclusively and partially breastfed infants are not calculated due to insufficient data (only one single spot sample of breast milk).

**Table 3 nutrients-13-01826-t003:** Non-parametric regression model of the univariate association between maternal and infant characteristics and infant urinary iodine concentration (µg/L) (*n* = 130). Numbers are presented as β with 95% confidence interval.

Independent Variables	ß and 95% CI ^a^	
Maternal age in Years	3.7	−1.2, 9.3
Maternal educational level		
<12 years	Reference	
12 years	0.0	−1.3, 0.9
1–4 years college/university	0.2	−1.9, 2.4
>4 years college/university	0.3	−1.6, 2.1
Maternal BMI, kg/m^2^		
18.5–24.9 (Normal weight)	Reference	
<18.5 (Underweight)	3.6	−2.1, 9.4
25–29.9 (Overweight)	−1.2	−10.4, 7.8
>30 (Obese)	1.9	−3.7, 6.8
Maternal iodine knowledge score		
Poor (0–5)	Reference	
Low (6–11)	4.7	−2.1, 13.8
Medium (12–19)	−3.2	−10.2, 5.4
High (20–26)	4.7	−2.1, 10.8
Maternal use of supplement last 24 h, yes	−0.3	−7.7, 6.9
Infant age in weeks	1.4	−0.2, 2.7
Infant gender, boy	3.1	−7.7, 13.2
Breastfeeding status		
Weaned	Reference	
Partially	−70.2	−121.8, −24.5
Exclusively	−79.6	−141.6, −36.5

^a^ CI = Confidence interval.

## Data Availability

The data presented in this study are available on request from the corresponding author. In order to meet ethical requirements for the use of confidential patient data, requests must be approved by the Regional Committee for Medical and Health Research Ethics in Norway.

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
