# Peer review of "Adequate Urinary Iodine Concentration among Infants in the Inland Area of Norway"

_nutrients, 2021, doi:10.3390/nu13061826_

Round 1

Reviewer 1 Report

Dear colleagues, your manuscript reflects a very well designed and carried out study, and the text is clear and complete. However, important messages are not strong enough, as for example the need to review the current recommendation of the median UIC for infants and young children using biomarkers of iodine status, and the difficulty to estimate dietary iodine intake. Thus, please consider the following comments for making your paper stronger and more influential:

  1. Line 19: Abstract I would suggest to delete the first sentence as this does not come from your study, and eliminating it does not reduce the importance of your paper.
  2. Lines 25-27: Abstract: Add more information about results of the estimated iodine intakes in the three different groups of children, and saying that these values are withing the current dietary recommendations of iodine for this age group in Norway. The same (results for each one of the three groups) should be done for the median UIC values.
  3. Line 27: Abstract: Delete “ranging from 15-679 ug/L” as the criterion for UIC is a median and not a cut-off point. Moreover, you have already presented the IQR values.
  4. Lines 27-29: Abstract: replace the current sentences saying that the UIC in weaned children was higher than in exclusively or partially breastfed children despite that the estimated dietary iodine was lower, and which suggest that estimation of dietary iodine may have not been accurate.
  5. Line 33: Abstract: Conclude with a recommendation about the need to review the current UIC criteria to estimate appropriate iodine intake in infants and young children based on association with metabolic biomarkers of iodine status.
  6. Line 45: Replace the current sentence for one that expresses better the conclusions of reference (5). Something like this make work: “However, a recent review concluded that there is insufficient good-quality evidence to support current recommendations for iodine supplementation in areas of mild- or moderate-deficiency of iodine”.
  7. Line 50: Add the words “intake and probably” before the word “status”.
  8. Line 67: Would it be possible to add a sentence specifying the technical basis why the Nordic countries recommend lower iodine intakes for young children that the WHO? I think that the Nordic recommendation might have stronger evidence than the WHO recommendation that is based on old information based only on the iodine content in breastmilk of a very few well-nourished mothers. If you do not have this information, it is fine to keep the text at it is now.
  9. Lines 221-228 and Table 2. Add sentences warning the reader that the apparent conflictive data in Table 2 are going to be analyzed in the discussion section. For example, if the weaned children have lower total 24-h iodine intake why the UIC was higher? Was this due to a lower volume of urine or to incorrect contents of iodine in the foods as presented in the food composition tables? Moreover, why the difference between habitual iodine intake and the 24-h intake in the partially breastfed children? In the methodology section you explained that for these children the iodine intake from breastmilk was not considered for estimating the habitual iodine intake, but the table does not include in a footnote this important methodological detail.
  10. Lines 234-237: The current recommendation of UIC (100-199 ug/L) refers to the median of the population; it is not a range of a minimum and a maximum threshold. Therefore, the percent of children below or above this range is meaningless. The results of UIC in table 2 shows that for these population, and using this criterion, exclusive and partial breast-fed children had an appropriate iodine intake, while those who have been weaned was slightly high. Thus, these sentences should be modified to be in agreement with the current use of the recommendation.
  11. Line 238: Your current sentence says that the association between breastfeeding status and UIC was significant. However, it does not specify how. If I understood your data correctly, at lower breastfeeding practice the higher the UIC. Am I right? However, this is not explained by the estimated iodine intake of the weaned children, which make me think that either the urinary volume was lower in these children or that there was an underestimation in the calculation of iodine intake through the diet. Some discussion to clarify this situation is needed.
  12. Figure 2: The last sentence should be corrected. “The red line shows the recommended median value for sufficient iodine concentration in urine (> 100 ug/L)”.
  13. Line 255: Perhaps adding a sentence that the lack of correlation between infant UIC and the estimated 24-hours dietary iodine intake may be evidence of inaccuracies in the estimation of the dietary iodine intake.
  14. Line 263: Replace twice the word “cut-off” with “median”.
  15. Line 267: Your sentence of disagreement with the WHO recommendation is very important and deserves additional explanation. I would suggest that you should propose that the current UIC-recommendation by WHO in small children must be reviewed using associations with biomarkers of iodine status, as the actual one may not reflect the true metabolic iodine status in infants and young children.
  16. Line 269: Although your statement is correct for exclusive breastfed children, it is more difficult to maintain for estimation of iodine intake from the diet. A note of caution may be good to add here.
  17. Lines 274—276: Although reference [24] is not yours, it needs some clarification in your text. Weaned infants may have less iodine intake coming from salt consumed by their mothers through the breastmilk, but not salt eaten directly by them. What about salt intake in complementary foods? You need to make these clarifications.
  18. Line 291: You say that “only one variable” was associated with infant UIC. You need to say which one: “No breastfeeding?”
  19. Lines 301 and 302: Replace the word “cut-off” with “median”
  20. Line 337: Why is not possible to use the iodine content in breastmilk estimated in two casual samples. Are the values of foods in composition tables more accurate that this?
  21. Lines 348-351: Place in past tense to differentiate clearly from your study, the last sentence of the paragraph.
  22. Line 355: I would suggest to change the order of the sentence as something like this: “Weaned infants had higher UIC than breastfed infants suggesting early access and consumption of other sources of iodine in addition to breastmilk”.
  23. Line 357: I would like to suggest to conclude with an additional sentence recommending the need to review the UIC recommendation for infants and young children based on biomarkers of iodine status.

Author Response

Thank you for all the valuable comments.

Please see the attachment for a point-by-point response.

Reviewer 2 Report

Thank you for giving me the opportunity to review the manuscript entitled “Adequate Urinary Iodine Concentration among Infants in the Inland Area of Norway”.

Please find below the specific comments to the manuscript:

  • This study (in the form as it is presented) seems to have rather local impact of the presented results
  • The authors might consider to change the title of the manuscript to correspond better to the results from the study. Similar is related to the aim of the study – it can benefit from extension (as in the paper more was done than only that stated in the aim).
  • The first part of the Methods (1. Study population) should clearly present the study sample (also with the data about refusals and excluded from the study) – that is presented as the first part of the results section but should be moved into the methods (figure 1 and line 203-209).
  • Strengths and limitations of the study needs to be clearly presented

Author Response

(The authors gave the same response as above.)

Round 2

Reviewer 2 Report

The authors have corrected the manuscript according to my comments.